# The Integration of Metabolomics, Electronic Tongue, and Chromatic Difference Reveals the Correlations between the Critical Compounds and Flavor Characteristics of Two Grades of High-Quality Dianhong Congou Black Tea

**DOI:** 10.3390/metabo13070864

**Published:** 2023-07-20

**Authors:** Shan Zhang, Xujiang Shan, Linchi Niu, Le Chen, Jinjin Wang, Qinghua Zhou, Haibo Yuan, Jia Li, Tian Wu

**Affiliations:** 1School of Landscape Architecture and Horticulture Sciences, Southwest Forestry University, Kunming 650224, China; zhangshan@tricaas.com; 2Key Laboratory of Tea Biology and Resources Utilization, Ministry of Agriculture, Tea Research Institute, Chinese Academy of Agricultural Sciences, Hangzhou 310008, China; shanxujiang@tricaas.com (X.S.); niulinchi2001@163.com (L.N.); yjscl@hotmail.com (L.C.); jinjinwangtkzc@tricaas.com (J.W.); 3State Key Laboratory of Tea Plant Biology and Utilization, Anhui Agricultural University, Hefei 230036, China; 4College of Environment, Zhejiang University of Technology, Hangzhou 310014, China; qhzhou@zjut.edu.cn

**Keywords:** metabolomics, electronic tongue, flavor, taste, Dianhong Congou black tea, grades

## Abstract

Tea’s biochemical compounds and flavor quality vary depending on its grade ranking. Dianhong Congou black tea (DCT) is a unique tea category produced using the large-leaf tea varieties from Yunnan, China. To date, the flavor characteristics and critical components of two grades of high-quality DCT, single-bud-grade DCT (BDCT), and special-grade DCT (SDCT) manufactured mainly with single buds and buds with one leaf, respectively, are far from clear. Herein, comparisons of two grades were performed by the integration of human sensory evaluation, an electronic tongue, chromatic differences, the quantification of major components, and metabolomics. The BDCT possessed a brisk, umami taste and a brighter infusion color, while the SDCT presented a comprehensive taste and redder liquor color. Quantification analysis showed that the levels of total polyphenols, catechins, and theaflavins (TFs) were significantly higher in the BDCT. Fifty-six different key compounds were screened by metabolomics, including catechins, flavone/flavonol glycosides, amino acids, phenolic acids, etc. Correlation analysis revealed that the sensory features of the BDCT and SDCT were attributed to their higher contents of catechins, TFs, theogallin, digalloylglucose, and accumulations of thearubigins (TRs), flavone/flavonol glycosides, and soluble sugars, respectively. This report is the first to focus on the comprehensive evaluation of the biochemical compositions and sensory characteristics of two grades of high-quality DCT, advancing the understanding of DCT from a multi-dimensional perspective.

## 1. Introduction

Black tea, as a recognized healthy beverage, has a long consumption history in the world, and it accounts for nearly 78% of the global tea consumption [1]. According to the processing technology used, black tea can be divided into three types, i.e., broken black tea, Congou black tea, and Souchoug black tea [2]. Among these, Congou black tea is popularly favored by consumers due to its elegant appearance, mellow taste, and sweet aroma [3]. Dianhong Congou black tea (DCT), which is a strip-shaped black tea, is a famous, geographically recognized brand produced by using the large-leaf tea varieties (*Camellia sinensis* L. var. assamica) from Yunnan province in the southwest of China [4] through the elaborate manufacturing steps of withering, rolling, fermentation, drying, and refining. Due to the special plant species and processing technology used, DCT is rich in tea polyphenols such as flavan-3-ols (catechins) and their derivatives such as theaflavins (TFs), thearubigins (TRs), and theasinensins (TBs), as well as phenolic acids and flavone/flavonol glycosides, etc., the contents of which are generally higher than in tea manufactured by small-/medium-leaf tea varieties [5]. On account of its higher content of tea polyphenols along with the other compounds such as amino acids, organic acids, and soluble sugars, DCT presents a strong, sweet-mellow, and umami taste profile in addition to its multiple health-protecting benefits [6,7]. In recent years, as an outstanding representative of premium Congou black tea, DCT has emerged as a popular herbal healthy beverage source and is exported to more than 30 countries around the world [5].

Grading is an important indicator for tea production as well as being a reference for consumers to estimate the tea quality. Based on the Group Standard of China Tea Science Society (T/CTSS 38-2021), DCT is divided into six grades, i.e., single-bud-, special-, first-, second-, third-, and fourth-grade, according to the tenderness of the young tea shoots, its sensory qualities, and its biochemical parameters, etc. Among these, single-bud-grade DCT (BDCT) and special-grade DCT (SDCT), which are produced by single buds with fresh leaf contents of no less than 90% and buds with one leaf, respectively, are commonly regarded as high-quality DCT according to the Group Standard of China Tea Science Society (T/CTSS 23-2021). Compared with middle-/low-quality DCT, high-quality DCT is favored by consumers, mostly due to its significant superiority in sensory characteristics [8]. However, it is difficult for average consumers to make a sensible choice between BDCT and SDCT, particularly due to the large gap in their prices in the market. In practice, consumers generally choose tea products based on the price, and more-expensive BDCT is often regarded as an optimal option. In fact, BDCT and SDCT both possess unique characteristics in terms of sensory quality and biochemical characterizations, which are suitable for various consumers’ preferences [8]. Therefore, it is necessary to elucidate the differences between the two grades of high-quality DCT in order to afford theoretical support to improve the productivity of high-quality DCT and to provide a guide for the consumption of high-quality black tea in a rational manner.

At present, numerous studies have focused on the selection of tea varieties, processing technology improvement, chemical composition variation, and evaluating the flavor quality of black tea. For instance, the ratio of total polyphenols/total amino acids (P/A value) is often used as an indicator for the prediction of the manufacturing suitability of a variety [9] and as an index for the evaluation of black tea quality [10]. The ratio of TFs/TRs is also an important index to measure the processing technology and flavor characteristics of black tea [11]. In addition, there are some studies centered on the flavor features [5], aroma patterns [8], and quality evaluation methods [12,13] of various grades of DCT. However, few studies have been conducted to investigate the sensory and molecular differences between the two grades of high-quality DCT, i.e., BDCT and SDCT.

Currently, the sensory assessment approaches include human sensory evaluation, electronic tongue analysis, and chromatic difference measurement, etc. [4]. The analytical tools for chemical analysis to determine the biochemical compounds of tea include high-performance liquid chromatography (HPLC), gas chromatography–mass spectrometry (GC-MS), and high-resolution liquid chromatography–mass spectrometry (LC-MS), etc. HPLC is an effective analytical technique for the analysis of tea’s major components. GC-MS and LC-MS are commonly performed for the qualitative and quantitative analysis of volatile and non-volatile compounds in tea in an untargeted pattern, respectively. A comprehensive combination of multiple sensory assessment approaches and chemical component analytical techniques is considered to be a more accurate method for the evaluation of tea quality. For example, Ma et al. characterized the key aroma-active compounds in high-grade Dianhong tea using GC-MS combined with sensory-directed flavor analysis [8]. Previous studies estimated the flavor quality of Keemun Congou black tea by a combination of sensory evaluation, HPLC, and LC-MS [14,15]. Therefore, it is advantageous to comprehensively study the two grades of high-quality DCT by integrating multiple approaches.

Herein, the aim of this study was to systematically compare the flavor characteristics of two grades of high-quality DCT, BDCT and SDCT, and to explore their correlations with the different key non-volatile compounds. To this end, a comprehensive sensory and molecular characterization on nine typical BDCT and nine typical SDCT samples was conducted by applying human sensory evaluation, an electronic tongue, chromatic differences, the quantification of main chemical components of the tea, and untargeted metabolomics profiling analysis. This study focused on the comparative assessment of two grades of high-quality DCT for the first time, intending to broaden the understanding about high-quality DCT with an objective, scientific, and global overview.

## 2. Materials and Methods

### 2.1. Chemicals and Reagents

High-purity solvents, including methanol, acetonitrile, formic acid, and acetic acid, were produced by either the Merck Company (Darmstadt, Germany) or the Sigma-Aldrich Company (St. Louis, MO, USA). Diagnostic solutions of the electronic tongue system used in this study including hydrochloric acid, sodium chloride, and monosodium glutamate (analytical grade) were purchased from the Evensen Biotechnology Company (Tianjin, China). Ultra-pure water was obtained using a Milli-Q System (Millipore, MA, USA).

### 2.2. Tea Samples

The 18 high-quality DCT samples, including 9 BDCT samples (mainly single-bud) and 9 SDCT samples (mainly one bud and one leaf), were collected from Fengqing county, Lincang city, Yunnan province, which is recognized as the core producing area of DCT [7].

### 2.3. Human Sensory Evaluation

Sensory evaluation of two groups of high-quality DCT (9 BDCT and 9 SDCT) was conducted following the Chinese National Standard (GB/T 23776-2018) by a group of professional tea tasters containing 2 females and 3 males from the Tea Research Institute, Chinese Academy of Agricultural Sciences. First, every tea sample (200 g) was placed in a white square plate for evaluating the dry tea’s appearance. Next, each tea sample (3 g) was brewed using 150 mL of boiling pure water for 5 min in a clean porcelain cup, and then the tea infusion was filtered into a bowl for evaluating the liquor color, aroma, taste, and infused leaf, sequentially. The comments were given referring to the Tea Vocabulary for Sensory Evaluation (GB/T 14487-2017). The scoring was conducted using a 100-point grading system: total score (100%) = dry tea appearance (25%) + liquor color (10%) + aroma (25%) + taste (30%) + infused leaf appearance (10%). The tasters had no prior knowledge about the tea samples. All tea samples were served randomly. More detailed information is supplied in the Appendix A.

### 2.4. Electronic Tongue Measurement

An electronic tongue sensing system (α-Astree II, Alpha MOS company, Toulouse, France) was employed to acquire the taste fingerprints of the two groups (9 BDCT and 9 SDCT). The sensory array included a reference electrode made of Ag/AgCl and seven receptors of NMS, ANS, SCS, AHS, CTS, PKS, and CPS for discriminating the taste of umami, sweetness, bitterness, sourness, saltiness, and two comprehensive indexes, respectively. The preparation and detection methods of the tea infusions referred to an earlier study [3]. To ensure the accuracy of the data, every tea sample was brewed twice, as per the methods described in Section 2.3, and each tea infusion was measured four times. The average of eight repetitions was used as the final electronic tongue response.

### 2.5. Chromatic Difference Assessment

A colorimeter system (CM-5, Konica Minolta Investment Company, Shanghai, China) comprising indexes of L* (luminance), a*(+) (redness), a*(−) (greenness), b*(+) (yellowness), b*(−) (blueness), and C* (color saturation) was used to characterize the infusion color of the 18 tea samples (9 BDCT and 9 SDCT). The tea infusion was prepared and tested as previously described [3]. Briefly, every tea sample was brewed twice, and each tea infusion was measured three times by the colorimeter. Ultra-pure water was used as a blank. The final value was obtained from the average of six repetitions.

### 2.6. Quantitative Determination of the Major Tea Chemical Components

The levels of total free amino acids and total polyphenols in the 18 tea samples (9 BDCT and 9 SDCT) were determined following the Chinese National Standards of GB/T 8314-2013 and GB/8313-2018, respectively. The content of total soluble sugars was detected using the anthrone–sulfuric method [16]. The quantitative determination of caffeine, epigallocatechin gallate (EGCG), epigallocatechin (EGC), epicatechin gallate (ECG), epicatechin (EC), and catechin (C), was conducted using an HPLC system (Shimadzu, Kyoto, Japan). The content of total catechins was obtained by summing up the individual catechins of EGCG, EGC, ECG, EC, and C. The quantification of total TFs, TRs, and TBs was implemented using the systematic analysis reported in an early report [17]. Each sample was extracted and analyzed with three replicates.

### 2.7. Untargeted Metabolomics Based on LC-MS Analysis

The metabolomics analysis was conducted as per a previous study [18]. Briefly, the tea metabolites were extracted by adding 70% methanol (*v*/*v*) into finely ground tea powder. Each tea sample was extracted and analyzed with three replicates. An LC-MS run was conducted using an UHPLC apparatus (Dionex Ultimate 3000 system, Thermo Fisher, CA, USA) coupled to a Q Exactive Plus MS instrument (Thermo Fisher, CA, USA). LC separation was performed on an ACQUITY UPLC HSS T3 column (2.1 mm × 100 mm, 1.8 um, Waters, MA, USA) by gradient elution using 0.1% formic acid (*v*/*v*) in pure water and 0.1% formic acid (*v*/*v*) in acetonitrile as phase A and phase B, respectively. Negative modes were operated in the full-scan (*m/z* range of 100–1000) and HCD MS/MS modes. The capillary temperature, voltage, sheath gas flow, and auxiliary gas flow were set as 300 °C, 3.8 kV, 25 arb, and 5 arb, respectively. The quality control (QC) samples, prepared by mixing equal aliquots of all samples, were detected every eight injections during the whole run. The metabolite annotation was performed using database queries from the online systems of HMDB (https://hmdb.ca/ (accessed on 24 October 2022)) and Metline (https://metlin.scripps.edu (accessed on 6 November 2022)), MS/MS fragments, exact mass (within 5 ppm), retention time, and authentic standards validation.

### 2.8. Data Processing, Analysis, and Visualization

The raw data acquired from the LC-MS analysis were processed using the XCMS 3.4.1 R-package for generating a peak list containing the peak area intensity, charge-to-mass ratio (*m/z*), and retention time. The data pretreatment included normalization to the total ion intensity, the 80% rule, and QC evaluation, as previously described in [16]. The Mann–Whitney’ U nonparametric test was used for the analysis of statistical differences between the two groups (9 BDCT and 9 SDCT). The Mann–Whitney U test, Bartlett’s test of sphericity, and the Kaiser–Meyer–Olkin (KMO) test were performed using SPSS 26.0.0.0 (IBM, New York, NY, USA). The principal component analysis (PCA), partial least-squares analysis (PLS), partial least-squares discriminate analysis (PLS–DA), and the variable importance for the projection (VIP) plot were performed using SIMCA–P 14.1 (Umetrics, Umeå, Sweden). The heatmap analysis was performed using TBtools–Ⅱ (v1.120, Toolbox for Biologists, Guangzhou, China). The graphs of the box plot were visualized using GraphPad Prism (GraphPad Software, San Diego, CA, USA). The pathway analysis was mapped by referring to the MetaboAnalyst (http://www.metaboanalyst.ca (accessed on 14 April 2023)) and KEGG (https://www.kegg.jp/ (accessed on 16 April 2023)) websites.

## 3. Results and Discussion

### 3.1. Human Sensory Evaluation

The sensory evaluation of the two grades of high-quality DCT, i.e., BDCT and SDCT, was performed by a group of experienced tea tasters. The results of the sensory features, including the dry tea appearance, liquor color, aroma, tea taste, infused leaf, and total score, are exhibited in Table 1. Detailed comments about and the scores of each tea sample are shown in Appendix A. In terms of the dry tea appearance, the BDCT teas were tight and heavy, with a black bloom color and a golden and tippy appearance, while the SDCT teas were tight and heavy, bent, and with a black bloom color and a slightly golden and tippy appearance, which resulted in significant differences in the comments and scores (Figure 1A, Table 1). It was supposed that the tea made from a single bud was more likely to be shaped with a tight strip and present a more golden and tippy appearance, as compared with the tea made from a bud with one leaf. Regarding the other factors, the two groups had no significant differences in their scores of the liquor color, aroma, taste, and infused leaf, but they exhibited differentiation in the comments (Table 1, Appendix A). Specifically, most of the BDCT samples presented a brisk taste with an umami, fruity-like sourness taste (Appendix A) and a brighter liquor color (Figure 1B), while most of the SDCT samples presented a thick, mellow, and sweet taste (Appendix A) and a redder liquor color (Figure 1B). In a word, the BDCT and SDCT showed no significant differences in their overall sensory scores, but they presented corresponding special sensory characteristics. To achieve an improved characterization, a more objective approach using an electronic tongue and chromatic difference analysis combined with a chemical investigation by quantifying the main chemical components and with untargeted metabolomics was conducted next.

### 3.2. Electronic Tongue Profiles Measurement

The representative taste patterns, as characterized by an electronic tongue, of the BDCT and SDCT samples were visualized by a radar chart (Figure 1C). The BDCT tea samples showed an obvious umami response (NMS, 7.48) and a sourness response (AHS, 7.72). The SDCT tea samples exhibited a comprehensive taste including a strong sweetness (ANS, 6.66), an apparent bitterness (SCS, 6.52) integrated with umami (NMS, 5.57), a sourness (AHS, 5.38), and higher comprehensive index values of CPS (6.71) and PKS (6.56). An existing study has reported that the sourness, which is generally caused by organic acids, is responsible for the fruity-like taste in many foods [19]. A moderate sourness is thought to be conducive to a harmonious taste in black tea liquor, and an appropriate bitterness is beneficial to the mellow and thick sensation in tea infusions [20]. The results indicated that the BDCT infusion emerged as having an umami, fruity-like sour taste, while the SDCT infusion presented a complex and multi-dimensional taste formed by an appropriate interplay of sweetness, bitterness, umami, and sourness. The results of the electronic tongue analysis were consistent with the human sensory comments in terms of taste.

### 3.3. Chromatic Difference Assessment

The color of a tea infusion is an important factor that affects the sensory quality of black tea. Premium black tea presents a bright and red liquor color, which is widely favored by consumers. Generally speaking, some consumers tend to like a redder liquor color, while others prefer a brighter liquor color. As shown in Figure 1D, the tea infusion in the BDCT presented a higher value of L* (luminance) (*p* < 0.001), reflecting a brighter liquor color compared with the SDCT. The a*(+) (redness) value in the SDCT group was significantly elevated (*p* < 0.001), indicating a redder liquor color in the SDCT. The values of b*(+) (yellowness) and C* (color saturation) were higher in the SDCT group, but they were not statistically significant (*p* > 0.05). The results were in agreement with the comments on the liquor color obtained by the human sensory evaluation, suggesting that each group of DCT had its own unique color that could satisfy different consumers’ preferences.

### 3.4. Quantitative Determination of the Major Chemical Constituents

The total amount of polyphenols in tea, which mainly include flavan-3-ols, dimeric/polymeric catechins, flavonols and flavone/flavonol glycosides, and phenolic acids, accounts for 18~36% (*w*/*w*) of the dry weight of tea leaves [21]. The ratio of P/A has been regarded as an indicator for determining the suitability of tea cultivars [9] and an index for estimating the quality of black tea [10]. As shown in Table 2, the level of total polyphenols was significantly higher (BDCT vs. SDCT, fold change = 1.14, *p* < 0.05) and the total amino acid content was slightly lower (no significant difference) in the BDCT. Thus, a higher P/A value was observed in the BDCT compared with the SDCT (BDCT vs. SDCT, FC = 1.16, *p* < 0.05). A previous study has reported that the level of total polyphenols is generally negatively correlated with the maturity of young tea shoots [22], and the level of total amino acids reaches higher levels in moderate-maturity tea leaves such as those with one bud with one leaf [23]. It is believed that this result was largely due to the tenderness differences in the fresh leaves of the BDCT and SDCT.

Total catechins, comprising galloylated catechins (EGCG, ECG) and non-galloylated catechins (EGC, EC, and C), account for 70~80% (*w*/*w*) of the amount of tea polyphenols [21]. The contents of total catechins and the individual compounds (EGCG, ECG, EGC, and EC) in the BDCT were significantly higher (*p* < 0.05) compared with the SDCT (Table 2). The changes in the concentrations of total catechins, ECG, and EGCG corresponded to the different tenderness of the young tea shoots. Catechin contents are negatively correlated with the growth of young shoots, with buds having a higher amount than first leaves [24]. As the important flavoring substances in tea, catechins are reported to impart a puckering astringency and a rough sensation in black tea infusions [1], which is generally described as “briskness” by sensory comments [25]. Therefore, we speculated that the higher amount of catechins may have been the factor contributing to the brisk taste in the BDCT.

Dimeric/polymeric catechins generated from oxidative condensation, i.e., TFs, TRs, and TBs, are the critical taste-active and colored substances in black tea. TFs impart a puckering, rough, and astringent sensation in tea infusions and are usually associated with the briskness taste and bright orange-yellow color in black tea infusion [26]. TRs are thought to be responsible for the redness color of tea liquors and the sweet taste of black tea [9]. An appropriate concentration of TBs has a positive effect on tea infusions, giving them a slightly sweet sensation, while an excessive amount of TBs is prone to cause a tea infusion to have a faint taste and dull color [21]. In addition, a moderately higher ratio of TFs/TRs is usually used as an index to measure the color and taste properties of black tea [11]. In this study, the contents of TRs and TBs were higher in the SDCT, though there were no significant differences (*p* > 0.05) (Table 2). Conversely, the concentration of TFs was significantly higher in the BDCT (*p* < 0.001), which was 1.36 times higher compared to the SDCT (Table 2). Furthermore, the ratio of TFs/TRs was obviously higher in the BDCT compared to the SDCT (*p* < 0.001) (Table 2). These results suggested that the BDCT teas were generally superior in terms of the features of a bright liquor color and a briskness taste due to the abundant accumulation of TFs, while the SDCT teas had advantages in terms of a redder liquor color and a sweet mellow taste due to the higher contents of TRs and TBs. In addition, the levels of total soluble sugars and caffeine showed no significant differences between the two groups.

In short, the results of the analysis of the main chemical constituents of the teas were basically in accordance with the aforementioned flavor features of the BDCT and SDCT. However, the tea flavor was influenced by the complex interplay of various quality-active compounds and was not limited to the several major components. Therefore, an untargeted metabolomics analysis was further conducted to investigate the global range of metabolites in the two grades of high-quality DCT.

### 3.5. Comprehensive Nontargeted Metabolomics Analysis

To comprehensively unravel the metabolic features of the BDCT and SDCT, an LC-MS-based untargeted metabolomics analysis was conducted. A total of 3104 ions were obtained. A typical chromatogram of the total ions is shown in Appendix A. To guarantee the reliability and repeatability of the data, the normalized intensities of the detected ions with replicate extractions were evaluated. The metabolite ions in the replicate extractions exhibited a high coefficient with R2 = 0.99 (Appendix A). Bartlett’s test of sphericity showed a high significance (chi–squared estimate of 527,322.793, *p* < 0.001), and the KMO value was observed to be 0.975, showing that the data were suitable for the factor analysis by PCA. A non-supervised PCA model including the BDCT, SDCT, and QC samples was used for a straightforward and global overview (Figure 2A). The QC samples were closely centered. The results suggested the reliable reproducibility of the present metabolomics analysis. In addition, the two groups were evidently separated in the PCA score plot, indicating a notable difference in metabolites in the two grades of tea samples. Next, a supervised PLS–DA model (R2X = 66.6%, R2Y = 98.7%, Q2 = 97.4%) was established (Figure 2B), by which a more obvious distinction of the two grades was gained. The cross-validation using 200 permutation tests with the R2 (0.0, 0.421) and Q2 (0.0, −0.309) intercepts demonstrated that the PLS–DA model was reliable (Figure 2C). Subsequently, an S-plot, which visualized the metabolites and the classification patterns in a covariance matrix, was generated (Figure 2D). The potential critical metabolites with important contributions to the classification are highlighted with red squares.

### 3.6. The Key Metabolic Characteristics

A total of 56 prominently different compounds between the BDCT and SDCT were screened using the Mann–Whitney U test (*p* < 0.05) and VIP value (VIP > 1), and they consisted of three flavan-3-ols, six catechin dimers, twenty-six flavonols and flavone/flavonol glycosides, seven phenolic acids, six amino acids, four sugars, two organic acids, one flavone, and one nucleotide. Information about these compounds, including their ionization, *m/z*, RT, *p* value, VIP value, and MS/MS fragmentation, are shown in Table 3. A heat map was used to visualize these different compounds in the BDCT and SDCT (Figure 3). The yellow and blue color in the color scale represent the metabolite occurring at a higher or lower level than the average level of all the tea samples. Notable differences in the two grades of high-quality DCT were revealed. Generally speaking, catechins, dimeric/trimeric catechins including theasinesins, procyanidins, and theaflavin-3,3-gallate (TF-3,3′-G), and a few phenolic acids with a galloyl group such as digalloylglucose, theogallin, etc., occurred at higher levels in the BDCT, while flavonols, flavone-*C*-glycosides, organic acids, soluble sugars, and most of the flavonol-*O*-glycosides, amino acids, and phenolic acids were at higher levels in the SDCT. To further elucidate the metabolite changes in the two grades of high-quality DCT manufactured with leaves with a different tenderness, the metabolic pathway involved with citric acid cycle (TCA cycle), phenylpropanoid metabolism, amino acid metabolism, flavone and flavonol metabolism, and flavonoid metabolism were mapped. The dynamic variations in the representative compounds between the two grades of high-quality DCT are shown in Figure 4.

#### 3.6.1. Flavan-3-ols and Their Derivatives

Flavan-3-ols, a group of the most abundant and characteristic metabolites in tea, have attracted much attention in biomedicine and food science due to their potent health-beneficial properties and are considered as the main contributors to the puckering astringency sensation in tea [1,27]. Among these, EGCG is the most abundant component affecting the taste of tea infusions and their bioactivity. An excessive intake of EGCG (734 mg/person/day) from green-tea-extract products is suspected to be related with liver toxicity, but daily tea consumption is considered to be safe since its EGCG content is much lower than the safe limit [28]. As shown in Figure 4, EGCG and ECG showed significantly higher levels in the BDCT than in the SDCT, which was in agreement with the results of the HPLC analysis (Table 2). Since ECG and EGCG belong to the group of galloylated catechins, this result was presumed to be possibly related with the higher galloylation level in buds than in leaves [29]. In addition, the downstream catechin derivatives, including procyanidins B1, procyanidins C1, theasinesins A, theasinesins B, theasinesins F, and TF-3,3′-G, also occurred at evidently higher levels in the BDCT (Figure 4). During black tea manufacturing, catechins undergo enzymatic oxidation to form various dimeric and oligomeric catechins, such as theasinesins, procyanidins, TFs, TRs, TBs, etc. [30]. Both catechins and their water-soluble oxidation products have been reported as being the main contributors to the taste sensation of tea liquor [21]. A moderately higher content of theasinesins has been considered as a characteristic of high-quality black tea [19]. TF-3,3′-G, which is formed by the condensation of ECG and EGCG, is a major component of TFs and confers a briskness taste in tea [31]. It was speculated that the sufficient substrates of ECG and EGCG might have been responsible for the higher content of TF-3,3′-G in the BDCT tea samples [14,31]. The content of TF-3,3′-G was significantly higher in the BDCT, which was consistent with the trend of the TF content described in Section 3.4. Hence, we speculated that the higher contents of EGCG, ECG, theasinesins A, theasinesins B, theasinesins F, and TF-3,3′-G might be the important factors causing the briskness taste in the BDCT tea infusion and that the accumulation of these compounds came from the corresponding sufficient substrate supply in the tea buds. In addition, TFs largely contribute to the brightness of tea liquid color [3]. In the metabolomics analysis, TF-3,3′-G accumulated significantly, which might have been one of the factors responsible for high value of L* (luminance) in the BDCT, as mentioned above.

#### 3.6.2. Phenolic Acids, Flavonols and Flavone/Flavonol Glycosides

Phenolic acids, as the precursors for the synthesis of catechins and flavonol glycosides, are important phenolic constituents with an antioxidative ability and contribute to the sourness, bitterness, and astringency taste in tea [27]. The representative different key compounds of phenolic acids, flavonols, and flavone/flavonol glycosides are exhibited in the metabolic pathway (Figure 4). Quinic acid and *p*-coumaric acid have been reported to be mainly responsible for the bitterness and astringency taste in tea liquid [32]. Theogallin, as a derivative of quinic acid, can enhance the umami taste in tea infusions [32]. As one of the hydrolysable tannins, digalloylglucose is considered to be correlated with the umami taste and higher quality of tea [33]. As manifested in Figure 4, *p*-coumaric acid and quinic acid demonstrated significantly higher levels in the SDCT and were beneficial in strengthening the bitterness and astringency taste in the tea infusions. On the contrary, the contents of theogallin and digalloylglucose were markedly higher in the BDCT, contributing to the umami taste and overall quality of the teas made from BDCT.

Flavonols and flavonol/flavone glycosides are also the main taste-active compounds in tea and present potential bioactivity in terms of antioxidant and cardiovascular-protective effects [27]. According to aglycones, flavonols and flavonol/flavone glycosides can be classified as apigenin-*C*-glycosides (ACGs), myricetin-*O*-glycosides (MOGs), quercetin-*O*-glycosides (QOGs), and kaempferol-*O*-glycosides (KOGs) [3]. As the downstream phenolic metabolites in the metabolic pathway, the contents of ACGs, MOGs, QOGs, and most of the KOGs were significantly higher in the SDCT than in the BDCT (Figure 4). These results were consistent with a previous report that showed that flavonol glycosides accumulated significantly more in White Peony tea (white tea processed using one bud with one leaf or two leaves) than in Silver Needle tea (white tea processed using only buds) [29]. As the second major phenolic metabolites in tea, flavonols and flavonol/flavone glycosides are mainly responsible for a mouth-drying astringency taste with an extremely low astringency taste threshold (0.001~19.8 µmol/L) [34], and they are responsible for the yellow liquor color of tea infusions [32]. In addition, it has been reported that flavonol glycosides can enhance the bitterness of caffeine [31]. As shown in Section 3.4, there were no significant differences in the content of caffeine between the two groups. However, the higher amount of flavonol glycosides was thought to enhance the bitterness taste of caffeine in the SDCT tea infusions. Therefore, the SDCT teas exhibited a higher intensity of SCS (bitterness), as reveled by the electronic tongue analysis (Figure 1C). Meanwhile, the higher value of b*(+) (yellowness) in the SDCT was considered to be related the yellowish color of flavonol glycosides (Figure 1D).

#### 3.6.3. Soluble Sugars, Amino Acids, and Organic Acids

Soluble sugars are largely responsible for the sweetness in tea [1]. In this study, the mono-, di-, and oligosaccharides in the tea liquors, i.e., glucose, maltose, and raffinose, were in higher levels in the SDCT (Figure 4), which was thought to be related with the sweet taste in the SDCT.

Amino acids, as a group of important taste-active compound species of tea, can be divided into three types (i.e., bitter-, umami-, and sweet-tasting amino acids) according to their taste features [35]. The contents of bitter-tasting amino acids, such as phenylalanine and tyrosine, were significantly retained in the SDCT (Figure 4), which might have strengthened the bitterness taste of the tea infusions in this group. Meanwhile, umami-tasting amino acids, such as glutamine, aspartic acid, and theanine, also showed upward trends in the SDCT. Glutamine and aspartic acid are the main contributors to the umami taste in tea infusions [35]. Theanine exhibits a sweetness or freshness taste at different concentrations [1]. As shown in Figure 4, the significantly higher concentrations of glutamine, aspartic acid, and theanine in the SDCT were thought to largely enhance the umami taste in the SDCT.

Organic acids, as the crucial intermediate compounds of the TCA cycle and the phenylpropanoid metabolism pathway, contribute the most to the acidity of tea, which is often described as a “fruity-like” taste in black tea when the acidity degree is appropriate [19]. Succinic acid and citric acid are considered to be the highest contributors to the acidity among the organic acids in black tea [20]. As has been previously reported, high responses of bitterness and astringency suppress the sour taste in tea, which is thought to be beneficial to the overall taste of black tea [20]. As shown in Figure 4, the levels of the acidic compounds of citric acid and succinic acid were significantly higher in the SDCT than in the BDCT. However, the higher responses of bitterness and astringency in the SDCT might have suppressed its sour taste. Therefore, the response of AHS (sourness) was much lower in the SDCT compared to in the BDCT.

In summary, the BDCT teas showed a briskness, an umami, fruity-like taste, and a brighter liquor color due to the accumulation of EGCG, ECG, TF-3,3′-G, theasinesins, theogallin, digalloylglucose, etc. Meanwhile, the SDCT teas presented a redder liquor color and a multi-dimensional flavor integrated with an interplay of a moderate sweetness, bitterness, umami, and sourness due to the higher contents of soluble sugars (glucose, maltose, and raffinose), phenolic acids (*p*-coumaric acid and quinic acid), flavonol glycosides (most KOGs, ACGs, MOGs, and QOGs), umami-tasting amino acids (glutamine, aspartic acid, and theanine), bitter-tasting amino acids (phenylalanine and tyrosine), and organic acids (succinic acid and citric acid).

### 3.7. Correlation Analysis between the Different Key Metabolites and the Sensory Indicators

Aiming to further elucidate the correlations between the different key metabolites and the sensory indicators of tea, a PLS analysis was performed (Figure 5). In the figure, the further a metabolite was suited from the original point, the greater its contribution to the sensory variation. The X-variables represent the intensities of the compounds, while the Y-variables indicate the strength of the taste or color indicators of the tea infusion. Variables being located in the same region suggested a close positive correlation among them. As shown in Figure 5, higher contents of succinic acid, ribonic acid, and UMP were thought to be strongly related with the greater intensities of NMS (umami) and AHS (sourness). In addition, the total catechins, the flavan-3-ols of EGCG and ECG, the polymerized catechins of procyanidin B1, procyanidin C1, theasinensin F, and theasinensin B, and the phenolic acid of theogallin were found to be positively correlated with the strength of L* (luminance), which was evidently observed in the infusions of the BDCT teas. In contrast, most of the flavonols and flavonol/flavone glycosides (particularly ACGs and KOGs) and caffeine showed positive correlations with the sensory indicators of SCS (bitterness). Furthermore, TRs and TBs largely contributed to the intensities of ANS (sweetness), CPS and PKS (comprehensive sensory index), a*(+) (redness), b*(+) (yellowness), and C* (color saturation), which were obviously noticed in the SDCT teas. These compounds were considered to have contributed to the respective taste and liquor color quality features of the BDCT and SDCT.

## 4. Conclusions

In this study, the flavor characteristics and potential critical compounds of two grades of high-quality DCT, BDCT and SDCT, were revealed by using human sensory evaluation, an electronic tongue, chromatic differences, the quantification of the main components of the teas, and untargeted metabolomic analysis. The BDCT possessed an apparent briskness, an umami, fruity-like taste sensation, and a brighter infusion color, while the SDCT presented a multi-dimensional taste integrated with a moderate sweetness, bitterness, umami, and sourness and a redder tea infusion color. Flavan-3-ols of EGCG and ECG, polymerized catechins of theasinensins, TFs, TRs, and TBs, flavonols and flavone/flavonol glycosides of ACGs, KOGs, and the phenolic acid of theogallin, etc., were the compounds that contributed the most to the flavor characteristics of the two grades of high-quality DCT. To our knowledge, this is the first report focusing on the correlations between the biochemical compositions and sensory characteristics of two grades of high-quality DCT. These comprehensive comparisons are expected to provide an objective identification basis and a scientific guide for consumers’ choices of high-quality DCT.

## Figures and Tables

**Figure 1 metabolites-13-00864-f001:**
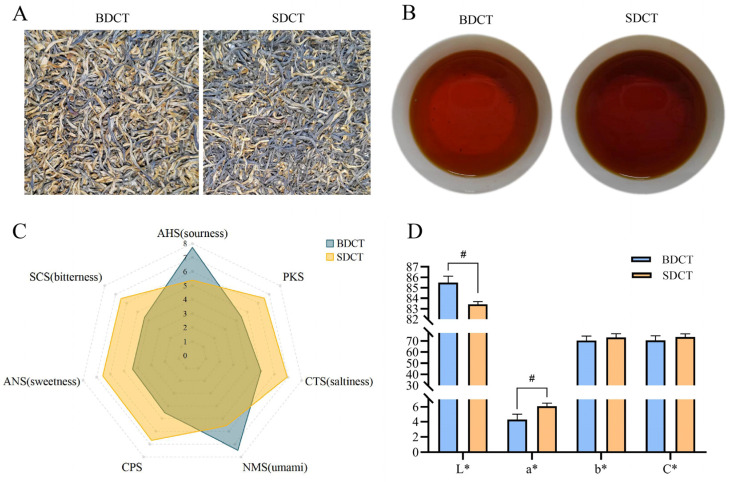
The representative tea samples’ dry tea appearances (**A**), infusion colors (**B**), taste intensities, as evaluated by electronic tongue (**C**), and bar plots of color attributes, as evaluated by chromatic difference measurement (**D**), of the two grades of high-quality DCT. # Indicates mean values with significant differences between the two groups (*p* < 0.05).

**Figure 2 metabolites-13-00864-f002:**
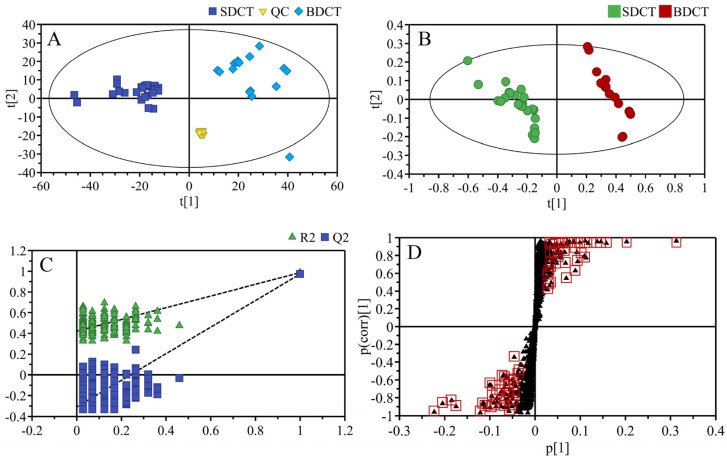
Multivariate statistical analysis of PCA score plot (**A**), PLS–DA score plot (**B**), cross-validation plot of PLS–DA model with 200 permutations (**C**), and S–plot of PLS−DA (**D**) for the two grades of high-quality DCT. The black triangles with red squares represent the potential important compounds.

**Figure 3 metabolites-13-00864-f003:**
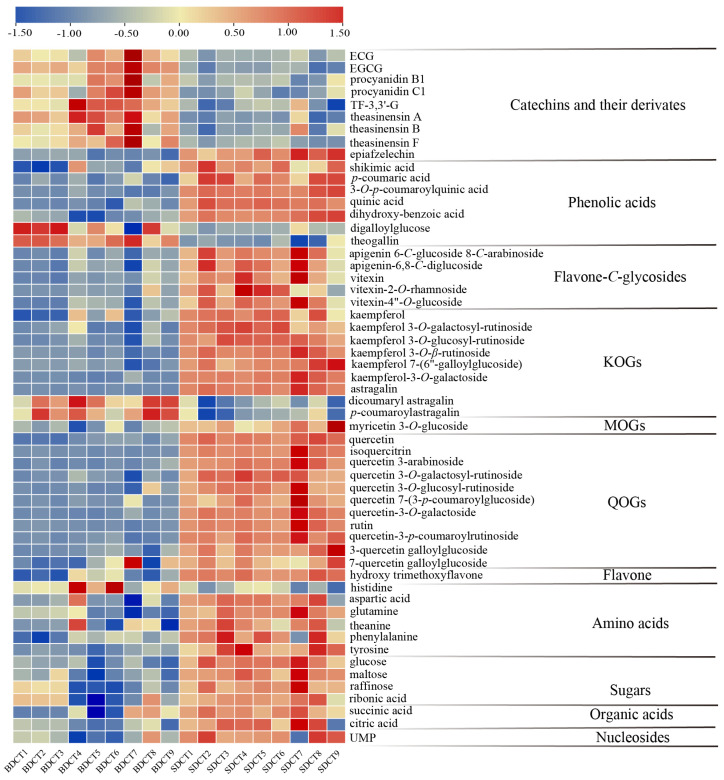
Heat map of the 56 different key compounds in the two grades of high-quality DCT. The data are shown as the mean of relative intensities of three replicates after being UV–scaled.

**Figure 4 metabolites-13-00864-f004:**
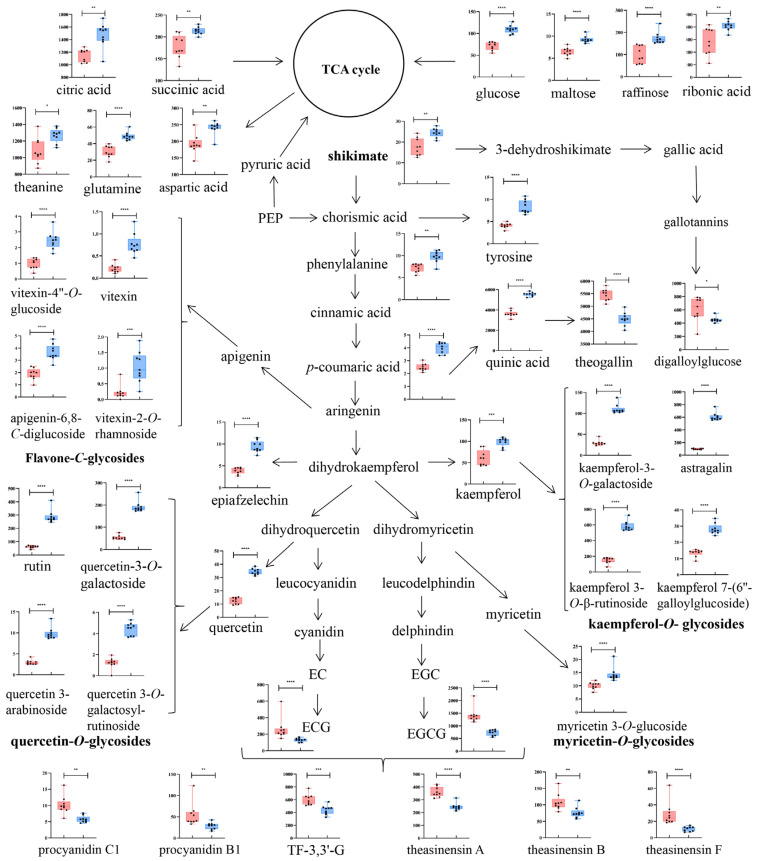
Mapping of the metabolic pathway and dynamic changes in the representative different key metabolites in two grades of high-quality DCT. Data are shown in scatter box plot as mean ± SD using relative abundance, as calculated by normalization to total ion intensity (×10^5^). The orange and blue boxes represent metabolite intensities in BDCT and SDCT, respectively. * indicates *p* < 0.05, ** indicates *p* < 0.001, *** represents *p* < 0.0005, **** suggests *p* < 0.0001. Statistical significance was determined by the Mann–Whitney U test.

**Figure 5 metabolites-13-00864-f005:**
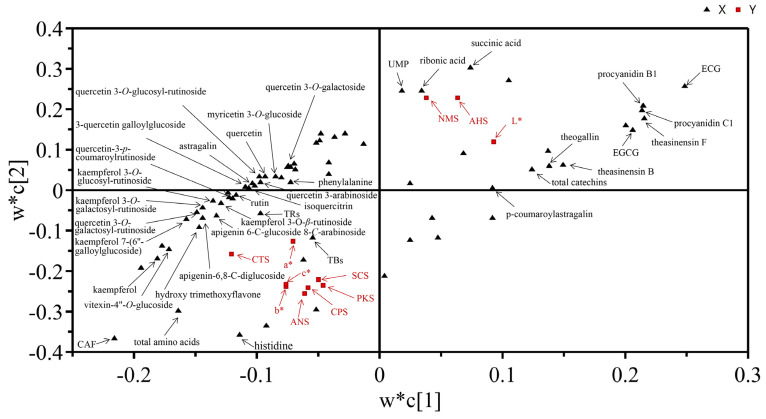
PLS analysis of the tea sensory indicators and the different key metabolites of the two grades of high-quality DCT. The metabolites were set as the X variables. The tea sensory indicators, including ANS, AHS, SCS, NMS, CTS, PKS, CPS, L*, a*, b*, and C*, were set as the Y variables.

**Table 1 metabolites-13-00864-t001:** Results of the sensory indicators (dry tea appearance, liquor color, aroma, tea taste, infused leaf appearance, and total score) of the two grades of high-quality DCT samples as evaluated by human sensory evaluation.

Group	Sample Number	Dry Tea Appearance (25%)	Liquor Color (10%)	Aroma (25%)	Tea Taste (30%)	Infused Leaf (10%)	Total Score (100%)
BDCT	9	96.33 ± 0.5 ^a^	90 ± 2.96	86.22 ± 4.82	84.56 ± 2.92	88.22 ± 1.64	88.83 ± 1.78
SDCT	9	93.56 ± 0.88 ^b^	88.89 ± 2.93	85.67 ± 3.54	85 ± 2.60	87.89 ± 2.62	87.98 ± 1.71

^a,b^ Different letters indicate significant differences between the mean scores of two groups (*p* < 0.05) as determined by the Mann–Whitney U test.

**Table 2 metabolites-13-00864-t002:** The contents of major biochemical components of tea in BDCT and SDCT.

Compounds	BDCT (*n* = 9 × 3)	SDCT (*n* = 9 × 3)	*p* Value	Fold Changes
Total polyphenols (%)	15.66 ± 2.22	13.62 ± 0.71	<0.05	1.14
Total amino acids (%)	3.37 ± 0.16	3.45 ± 0.18	n. s.	0.97
Total soluble sugars (%)	5.90 ± 0.46	6.09 ± 0.48	n. s.	0.99
Caffeine (%)	1.46 ± 0.42	1.72 ± 0.24	n. s.	0.88
Total catechins (%)	12.23 ± 1.95	9.20 ± 1.31	<0.05	1.34
EGCG (%)	8.78 ± 1.43	6.82 ± 0.78	<0.05	1.29
ECG (%)	1.52 ± 0.28	0.86 ± 0.18	<0.05	1.77
EGC (%)	1.30 ± 0.24	0.98 ± 0.17	<0.05	1.33
EC (%)	0.52 ± 0.22	0.30 ± 0.06	<0.05	1.76
C (%)	0.10 ± 0.06	0.13 ± 0.03	n. s.	0.83
TFs (%)	0.22 ± 0.05	0.17 ± 0.04	<0.001	1.36
TRs (%)	2.53 ± 0.32	2.64 ± 2.64	n. s.	0.96
TBs (%)	5.16 ± 0.70	5.30 ± 0.66	n. s.	1.03
TFs/TRs	0.09 ± 0.01	0.06 ± 0.01	<0.001	1.41
P/A value	4.64 ± 0.58	3.98 ± 0.36	<0.05	1.16

n. s., no significant difference; *p* < 0.05, significant difference; *p* < 0.001, extremely significant difference; the same below. Statistical significance was determined by the Mann–Whitney U test.

**Table 3 metabolites-13-00864-t003:** Detailed information of 56 key different compounds between BDCT and SDCT screened based on *p* < 0.05 and VIP > 1.

No.	Metabolite Identification	*m/z*	RT/min	*p* Value	VIP	MS/MS
Flavan-3-ols and their derivatives						
1	Epiafzelechin ^a^	273.0773	7.7	<0.001	1.6	187, 189, 229, 255
2	ECG ^a^	305.0665	5.1	<0.001	1.1	125, 137, 165, 179, 219, 221, 261, 287
3	EGCG ^a^	457.0767	6.6	<0.001	1.5	169, 193, 287, 305, 331
4	Procyanidin B1 ^a^	577.1351	5.2	<0.001	1.1	125, 289, 407, 425, 451, 559
5	Procyanidin C1 ^b^	865.1985	6.2	<0.001	1.2	125, 289, 407, 577, 695, 713, 739, 847
6	Theasinensin A ^b^	913.1469	5.8	<0.001	1.5	285, 423, 573, 591, 743, 761
7	Theasinensin B ^b^	761.1359	4.6	<0.001	1.2	423, 483, 575, 593, 609, 635, 743
8	Theasinensin F ^b^	897.1520	7.2	<0.001	1.3	407, 727, 745
9	TF-3,3′-G ^a^	867.1408	11.9	<0.001	1.3	125, 169, 241
Flavonols and flavone/flavonol glycosides						
10	Apigenin 6-*C*-glucoside 8-*C*-arabinoside ^b^	563.1406	7.5	<0.001	1.5	353, 383, 524, 443, 473, 503, 545
11	Apigenin-6,8-*C*-diglucoside ^b^	593.1512	6.5	<0.001	1.4	473, 353, 503, 383, 575
12	Vitexin ^a^	431.0983	8.6	<0.001	1.5	283, 311, 341
13	Vitexin-2-*O*-rhamnoside ^a^	577.1563	8.6	<0.001	1.1	413, 293, 457
14	Vitexin-4″-*O*-glucoside ^b^	593.1506	8.1	<0.001	1.5	293, 413
15	Kaempferol ^a^	285.0414	12.2	<0.001	1.4	227, 239, 211
16	Kaempferol 3-*O*-galactosyl-rutinoside ^b^	755.204	8.7	<0.001	1.4	285
17	Kaempferol 3-*O*-glucosyl-rutinoside ^b^	755.204	9.1	<0.001	1.6	285
18	Kaempferol 3-*O*-β-rutinoside ^b^	593.1506	9.7	<0.001	1.7	285, 327
19	Kaempferol 7-(6″-galloylglucoside) ^b^	599.1075	9.8	<0.001	1.6	125, 169, 313, 285, 447
20	Kaempferol-3-*O*-galactoside ^b^	447.0933	9.7	<0.001	1.7	255, 284, 285, 327, 357
21	Dicoumaryl astragalin ^b^	739.1675	12.2	<0.001	1.3	145, 285, 453, 593
22	*p*-Coumaroylastragalin ^b^	593.1306	11.9	<0.001	1.3	285, 307, 447
23	Astragalin ^a^	447.0933	10.2	<0.001	1.7	255, 284, 285, 327, 357
24	Myricetin 3-*O*-glucoside ^b^	479.0825	7.7	<0.001	1.3	316, 317, 271
25	Quercetin ^a^	301.0348	12.0	<0.001	1.6	107, 121, 151, 179
26	Isoquercitrin ^a^	463.0882	9.0	<0.001	1.7	301, 300
27	Quercetin 3-arabinoside ^b^	433.0799	9.7	<0.001	1.7	300, 271, 301, 255
28	Quercetin 3-*O*-galactosyl-rutinoside ^b^	771.1989	8.0	<0.001	1.6	301, 343, 609
29	Quercetin 3-*O*-glucosyl-rutinoside ^b^	771.1989	8.2	<0.001	1.5	301, 343, 609
30	Quercetin 7-(3-*p*-coumaroylglucoside) ^b^	609.1279	11.8	<0.001	1.5	463, 300, 301
31	Quercetin-3-*O*-galactoside ^b^	463.0882	8.8	<0.001	1.7	301, 300, 293
32	Quercetin-3-*p*-coumaroylrutinoside ^b^	755.1873	11.8	<0.001	1.6	609, 591, 301, 271
33	3-Quercetin galloylglucoside ^b^	615.1027	8.4	<0.001	1.4	463, 300, 301
34	7-Quercetin galloylglucoside ^b^	615.1027	8.4	<0.001	1.4	463, 300, 301
35	Rutin ^a^	609.1461	8.6	<0.001	1.7	301, 343
Amino acids						
36	Aspartic acid ^a^	132.0296	0.7	<0.001	1.3	88, 115
37	Glutamine ^b^	146.0453	0.7	<0.001	1.4	109, 127
38	Histidine ^a^	154.0616	0.6	<0.05	1.1	93, 137
39	Phenylalanine ^a^	164.0711	2.4	<0.001	1.3	97, 137, 147
40	Theanine ^a^	173.0926	1.1	<0.05	1.1	128, 155
41	Tyrosine ^a^	180.066	1.2	<0.001	1.4	72, 93, 119, 163
Phenolic acids						
42	Digalloylglucose ^b^	483.078	5.2	<0.05	1.1	125, 169, 211,271, 313, 331
43	Dihydroxy-benzoic acid ^b^	153.0182	6.2	<0.05	1.6	109
44	Quinic acid ^a^	191.0561	0.7	<0.001	1.6	85, 93, 127, 173
45	Shikimic acid ^a^	173.0455	0.8	<0.001	1.1	73, 93, 111, 137
46	Theogallin ^a^	343.0671	1.8	<0.001	1.5	191
47	*p*-Coumaric acid ^a^	163.04	5.2	<0.001	1.4	119, 93
48	3-*O*-*p*-coumaroylquinic acid ^b^	337.0929	6.2	<0.001	1.7	173
Sugars						
49	Glucose ^a^	179.0562	0.8	<0.001	1.6	59, 71, 89, 101, 113
50	Maltose ^a^	341.1089	0.8	<0.001	1.3	113, 119, 143, 161, 179
51	Raffinose ^a^	503.1612	0.7	<0.001	1.4	89, 101, 179, 221
52	ribonic acid ^b^	165.0398	0.7	<0.001	1.1	75, 105, 129, 147
Organic acids						
53	Citric acid ^a^	191.0197	1.1	<0.001	1.2	85, 111, 173
54	Succinic acid ^a^	117.0187	1.3	<0.001	1.1	73, 99
Flavone						
55	Hydroxy trimethoxyflavone ^b^	327.0893	8.0	<0.05	1.6	237, 211, 265
Nucleotide						
56	UMP ^b^	323.0286	0.8	<0.001	1.2	173, 211, 279, 305, 79, 193

^a^ Confirmed by standards. ^b^ Identified based on exact mass and MS/MS. Statistical significance was determined by the Mann–Whitney U test.

## Data Availability

The data presented in this study are available within the article and in the Appendix A.

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
