# Peer review of "The Integration of Metabolomics, Electronic Tongue, and Chromatic Difference Reveals the Correlations between the Critical Compounds and Flavor Characteristics of Two Grades of High-Quality Dianhong Congou Black Tea"

_metabolites, 2023, doi:10.3390/metabo13070864_

Round 1

Reviewer 1 Report

General comments

Authors describe and indicate the relation between flavor characteristics (human sensory evaluation, electronic tongue, chromatic difference) on one side and components quantification and un-targeted metabolomic analysis on the other side. The research aims to generate for the first time an objective identification basis of high-quality DCT.

Detailed comments

Abstract

L22: it would be helpful to shortly mention the difference between BDCT and SDCT

Introduction

L79:..as we know, still no research – please correct.

L82-90: please add some information about differences/advantages of using either of the mentioned techniques for aroma analysis.

Authors mention in “Results” the importance of the ratio of total polyphenols/total amino acids (P/A value) as an indicator for manufacturing suitability - please comment on that issue in the Introduction.

Materials and Methods

2.5-2.7: only two replicas were measured? – please comment. At least three measurements are recommended.

Concerning the techniques presented in the other paragraphs of MM, sample number is not indicated

Results and Discussion

Fig. 1 D: how was deviation and statistical differences evaluated based on two replicas?

L282: … the metabolites that profoundly relate to the taste and color features in two grades of high-quality DCT – please correct.

L348:… were also occurred at.. ; L349:… catechins undergo….L386:… As shown…L394:…that was supposed to be related….L401:… which often is described… -please correct.

Fig S2: normalized intensities (please correct axis title)

Table S1: gloden  - change to golden

Some minor English corrections are required in " Results and Discussion"

Reviewer 2 Report

After reviewing the manuscript titled: "Integration of metabolomics, electronic tongue and chromatic difference reveal the correlations between critical compounds and flavor characteristics of two grades of high-quality Dianhong Congou black tea" I have some comments and suggestions outlined below. 

Italicize Latin names.

Line 105 - I suggest replacing "Solvents in chromatographic grade" with "High purity solvents".

Line 117 - Only 5 tea testers seem could influence the reliability and generalizability of the results. How is this relevant?

Line 141 - Why only two replicates? How did you do the statistical analysis and reveal there were significant differences? Line 202 - on graph D, clearly there should be no significant differences between the samples, according to the presented. Which test was used to compare? Mann-Whitney U test?

It would be beneficial to know if differences in color would impact the consumer's perception and preference.

Line 297 - Please provide KMO value and Bartlett's Test of Sphericity for the PCA.

Can you clarify why the Mann-Whitney U test (p < 0.01) and VIP values (VIP > 1) were used?

Comment on the health implications (positive and negative) of chemical constituents differences, as this would be interesting to consumers and the scientific community alike.

Elaborate more on how specific compounds influence sensory attributes.

Clearer language and better presentation are necessary to enhance the readability and comprehension of the manuscript. Occasionally it is very hard to follow what authors are trying to convey.

Clearer language and better presentation are necessary to enhance the readability and comprehension of the manuscript.

Reviewer 3 Report

  The manuscript is well-written and coherent. Some remarks and questions:
  • latin name should be written in Italics
  • how was exactly the sensory test performed? 18 sample are too much in a single session. The relevant ISO norm specifies a maximum of 6 samples per session.
  • L 145: at least the principle of the measurement should be reported

English has to be checked.

Round 2

Reviewer 1 Report

Reviewer comments to the original manuscript were all addressed satisfactorily. 

In L77_83:please correct the newly added phrases - For instance, the ratio of total polyphenols/total amino acids (P/A value) is often be used as an indicator for the prediction of manufacturing suitability of varieties, Chen et al., 2022), either for the black tea quality evaluation (Dong, et al., 2017) – either/or – the sentence is missing something.

Some minor English corrections must be performed as indicated in "Comments and suggestions to authors"

Reviewer 2 Report

The authors substantially changed the manuscript according to the suggestions and comments. The manuscript can now be accepted for publication.

Minor editing of English language required
